# Mild Lactic Acid Stress Causes Strain-Dependent Reduction in SEC Protein Levels

**DOI:** 10.3390/microorganisms9051014

**Published:** 2021-05-08

**Authors:** Danai Etter, Céline Jenni, Taurai Tasara, Sophia Johler

**Affiliations:** 1Institute for Food Safety and Hygiene, University of Zurich, 8057 Zurich, Switzerland; danai.etter@uzh.ch (D.E.); taurai.tasara@uzh.ch (T.T.); 2Laboratory of Food Microbiology, Institute for Food, Nutrition and Health (IFNH), ETH Zurich, 8092 Zurich, Switzerland; jennic@student.ethz.ch

**Keywords:** superantigen, mastitis, food intoxication, regulation, sec variants

## Abstract

Staphylococcal enterotoxin C (SEC) is a major cause of staphylococcal food poisoning in humans and plays a role in bovine mastitis. *Staphylococcus aureus* (*S. aureus*) benefits from a competitive growth advantage under stress conditions encountered in foods such as a low pH. Therefore, understanding the role of stressors such as lactic acid on SEC production is of pivotal relevance to food safety. However, stress-dependent cues and their effects on enterotoxin expression are still poorly understood. In this study, we used human and animal strains harboring different SEC variants in order to evaluate the influence of mild lactic acid stress (pH 6.0) on SEC expression both on transcriptional and translational level. Although only a modest decrease in *sec* mRNA levels was observed under lactic acid stress, protein levels showed a significant decrease in SEC levels for some strains. These findings indicate that post-transcriptional modifications can act in SEC expression under lactic acid stress.

## 1. Introduction

*Staphylococcus aureus* (*S. aureus*) is of major relevance in food intoxications and infectious diseases of humans and animals [1,2]. *S. aureus* employs a plethora of virulence factors including secreted enterotoxins (SEs). They lead to an emetic response when ingested and act as superantigens [3,4]. The consumption of one or several preformed enterotoxins produced by *S. aureus* causes staphylococcal food poisoning (SFP). SFP symptoms include nausea, vomiting, and abdominal pain, followed by diarrhea [5]. It is one of the most common causes of foodborne intoxications worldwide [6]. The European Food Safety Authority (EFSA) reported 393 SFP outbreaks in 2014 and an increase in cases with 434 outbreaks in 2016 [7,8]. Moreover, in the United States, the Centers for Disease Control and Prevention (CDC) reported 17 SFP outbreaks and 566 cases in 2014. The most common contributing factors in these outbreaks were improper maintenance of the cold chain and inadequate food preparation practices, leading to the proliferation of pathogens and the concomitant production of enterotoxins [9].

Of the currently 25 known SEs, SEC is of particular interest since various, often host-specific variants have been reported [10,11,12,13]. SEC plays a crucial role in SFP and the development of atopic dermatitis [14]. SEC is also frequently found in milk and milk products [15,16,17,18,19,20,21,22,23,24] and represents a key driver of the inflammatory response in bovine mastitis in dairy cattle [25,26,27,28]. We recently provided a comprehensive review of SEC variants and their role in SFP, as well as their structure and properties [10].

To reduce the burden of *S. aureus* in the dairy value chain, lactic acid bacteria (LAB) have been used as an intervention in mastitis treatment and as starter cultures in cheese production. For instance, LAB were successfully used to alleviate mastitis symptoms in cows [29] and to inhibit mastitis-causing pathogens including *S. aureus* [30]. The use of *Weissella paramesenteroides* GIR16L4 and *Lactobacillus rhamnosus* D1 as starter cultures was shown to decrease SEC expression in several *S. aureus* strains [31]. Another study showed decreased expression of *sec* and SEC in co-cultures with LAB compared with pure *S. aureus* cultures by up to 331-fold in TSB and milk [32]. Metabolites of LAB, in particular lactic acid, might present an additional metabolic burden to *S. aureus* and therefore interfere with toxin expression.

Lactic acid is a lipophilic weak organic acid that freely dissociates through the bacterial membrane. Inside the cell, lactic acid dissociates and thereby releases protons that acidify the cytoplasm. Additional energy is required to maintain internal pH, leading to adaptations in bacterial metabolic processes [33,34]. It can therefore alter toxin regulation by interfering with *S. aureus* regulatory systems such as the accessory gene regulator (*agr*) [35]. The *agr* regulon employs a multicomponent system that is activated by autoinducing peptides (AIP). Upon activation, RNAIII is transcribed, which deactivates the repressor of toxins (*rot*) [36]. The activity of *agr* may be influenced by other regulatory elements that react to changes in the microenvironment or external stressors.

External stressors that have been shown to influence SE production include NaCl [37], low pH [38], nitrite [39], and others [40,41]. Although previous studies demonstrated the substantial resilience of *S. aureus* against low pH, with growth being observed at pH 4 [42,43,44], pH stress can still influence toxin expression. Lactic acid was previously shown to affect SEA and SED production [45,46,47]. In the presence of LAB, *agrA*, *sarA*, and *sigB* are typically downregulated, while *rot* is upregulated [48]. However, it remains unclear how external stressors affect the complex regulatory network and whether the presence of lactic acid influences SEC expression. Therefore, we investigated the role of lactic acid in SEC production on mRNA and protein level.

## 2. Materials and Methods

### 2.1. Bacterial Strains, Growth Conditions, and Sample Collection for sec mRNA and SEC Protein Quantification

All *S. aureus* strains used in this study are listed in Table 1. The strains were grown in LB medium (non-stress control conditions) and in LB supplemented with lactic acid. Mild acidic stress conditions encountered in food were mimicked by adjusting the LB medium (BD, France) to pH 6.0 using ~0.6 mL 90% (*v*/*v*) lactic acid (Merck, Darmstadt, Germany). The medium was buffered using 19.52 g 2-(N-Morpholino) ethanesulfonic acidhydrate (MES hydrate, Sigma-Aldrich, Switzerland) per 1000 mL LB. pH was monitored for 2 representative strains and remained unchanged over the course of the experiment (Appendix A). All media were sterile filtered and stored at 4 °C.

Single colonies from each strain were transferred from 5% sheep blood agar to 5 mL LB broth and grown overnight (37 °C, 125 rpm). Overnight cultures were centrifuged (5000× *g*, 2 min) and washed twice with 0.85% NaCl, before resuspending the pellet in 0.85% saline solution. We adjusted 50 mL of medium (LB and LB + lactic acid) to an OD_600_ of 0.05 using the washed bacteria. The culture was incubated at 37 °C at 125 rpm and harvested after 4, 10, and 24 h during exponential, early stationary, and late stationary growth phase, respectively. Three independent biological replicates were collected.

Growth curves were evaluated by plating serial dilutions on plate count agar (Oxoid, Pratteln, Switzerland) as previously described [37] with minor modifications. Namely, the culture was adjusted to a final OD of 0.05 in 50 mL medium in 250 mL Erlenmeyer flasks. Two independent replicates were assessed.

For *sec* mRNA quantification, 1 mL of sample was added to 3 mL RNAprotect^®®^Tissue Reagent (Qiagen, Hilden, Germany) and processed according to manufacturer’s instructions. The cell pellets were stored at −20 °C until further processing. For SEC protein quantification, 1 mL of sample was collected in low protein binding micro-centrifuge tubes (Thermo Scientific, Waltham, MA, USA) and stored at −20 °C until further processing.

### 2.2. RNA Extraction

RNA extraction was performed with the RNeasy mini Kit Plus (Qiagen, Hilden, Germany) as previously described [52]. All RNA samples were quantified using QuantusFluorometer (Promega, Dübendorf, Switzerland) instruments. Quality control was performed by the Agilent 2100 Bioanalyzer (Agilent Technologies, Waldbronn, Germany) instrument using Agilent RNA 6000 PicoReagents according to the manufacturer’s instructions. Samples were included in the study if they met the inclusion criteria of RNA integrity number > 6. RNA integrity numbers ranged from 6.18.5.

### 2.3. Reverse-Transcription and Quantitative Real-Time PCR

All RNA samples were diluted to 40 ng/μL in RNase-free water. A total of 480 ng was converted to cDNA using the QuantiTect^®®^Reverse Transcription Kit (Qiagen, Germany) according to the manufacturer’s instructions. A no-RT control and a negative control were included in every run. The final cDNA was diluted 1:10 with DNase-free water (Promega, Madison, WI, USA) and stored at −20 °C. The following primers sequences were used for real-time qPCR: forward 5′TAA CGG CAA TAC TTT TTG GT3′ and reverse primer 5′AGG TGG ACT TCT ATC TTC AC3′. A 5 μL template was added to 10 μL LightCycler^®®^ 480 SYBR Green I Master, 2 μL of each primer (5 μM), and 1 μL nuclease-free water in LightCycler^®®^480 Mulitwell Plate 96white (Roche, Basel, Germany). The plate was centrifuged for 2 min at 1500× *g*. Quantification was performed on the Lightcycler^®®^96 Instrument (Roche, Basel, Switzerland) as previously described [52]. The relative expression of the target gene *sec* was normalized using the housekeeping genes *rho* and *rplD* [52]. Ct values were determined using the Lightcycler^®®^Software v. 1.1.0.1320 (Roche). The influence of lactic acid stress on *sec* expression in each strain is expressed as Δct values and relative expression (2^-ΔΔct^). The following formula was used to calculate expression values: 2^-(ref control-sec control)-(ref lactic acid-sec lactic acid)^. Statistical analysis was performed with RStudio 1.3.1093 and GraphPad Prism 9.0.0. For RNA analysis a mixed-effect linear model was fitted on the fold change, with a full three-way interaction between reference gene, strain, and time effects. Fold change was log_10_-transformed to ensure normal distribution. To determine whether individual mRNA levels were increased or decreased (indicated by a fold change significantly larger than 1), lsmeans was used to perform a two-sided effect test, with Holm–Bonferroni-corrected *p*-values. The results were regarded as significant if *p* < 0.05.

### 2.4. Protein Quantification

An enzyme-linked immunosorbent assay (ELISA) was developed to quantify the effect of mild lactic acid stress on SEC protein levels. The protocol was based on [53] with some modifications according to [41]. Sheep Anti-SEC IgG (Toxin Technology Inc., Sarasota, FL, USA) was used to measure SEC concentrations. For quantification, a standard curve was obtained using SEC_2_ (Toxin Technology, Inc., USA). Absorbance was measured at 405 nm in a Synergy HT plate reader (BioTek, Sursee, Switzerland). Absorbance values were plotted against toxin concentrations, and values were determined from linear regression in Excel (version 16.44). ELISA measurements were performed in duplicates. Statistical analysis was performed with RStudio 1.3.1093 and GraphPad Prism 9.0.0. Protein data were analyzed via two-way ANOVA and post hoc Tukey’s multiple comparisons. Results were regarded as significant if *p* < 0.05.

## 3. Results

### 3.1. Effect of Mild Lactic Acid Stress (pH 6.0) on Bacterial Growth and sec mRNA Levels

The bacterial growth of the seven *S. aureus* strains was compared in LB and LB supplemented with lactic acid by plate counting. The growth behavior was similar for all strains under control and stress conditions. Growth was not impaired by lactic acid in any of the investigated strains (Appendix A).

Under lactic acid stress, we observed a trend toward decreased *sec* expression (Figure 1) compared with control conditions. However, the reduction in *sec* expression was strain-dependent, being only significant in strains BW10 (10 and 24 h), NB6 (24 h), SAR1 (4 h), SAR38 (4 and 10 h), and OV20 (10 h).

### 3.2. SEC Protein Levels under Lactic Acid Stress

SEC concentrations under lactic acid stress and control conditions were assessed by ELISA at 4, 10, and 24 h (Figure 2, Table 2). Under control conditions, two different expression levels were observed. BW10 and SAI48 were classified as SEC over-producers (>1000 ng/mL) with concentrations ranging from 3410 ng/mL to 9867 ng/mL after 24 h, respectively. NB6, SAI3, SAR1, SAR38, and OV20 were classified as low to moderate level SEC producers (<1000 ng/mL) with concentrations from 54.5 to 344.9 ng/mL after 24 h (Table 2). Expression levels were lowest after 4 h and highest after 24 h for all strains.

SEC concentrations under lactic acid stress were generally lower than those under non-stress control conditions, with the exception of SAI48, SAR1, and SAR38 (Figure 2). BW10 and SAI48 again showed the highest expression levels with 740 ± 69 and 4887 ± 1027 ng/mL at the late exponential phase, respectively. The low-to-moderate SEC producers NB6, SAI3, SAR1, SAR38, and OV20 ranged from 1 ng/mL (OV20) to 142 ng/mL (OV20) over all time points. SEC production in the human infection isolate SAI48 was the least impaired by lactic acid stress (−29%) while the ovine mastitis isolate OV20 was affected the most (−255%) (Table 2).

## 4. Discussion

Since *S. aureus* possesses a competitive growth advantage in many food matrices, it is crucial to identify compounds that interfere with SE production. Here, we used lactic acid and pH 6.0 to mimic conditions comparable to food matrices such as ham, cheese, and other fermented products [47]. LB medium was used to ensure reproducibility and to allow observation of the effect of lactic acid as an individual constituent. Experiments were performed at 37 °C to provide optimal growth conditions for *S. aureus*.

In the investigated strains, lactic acid stress resulted in a trend toward lower *sec* transcription, although the results were not significant for all strains. Significant reductions in mRNA levels were observed for strains isolated from food and bovine mastitis, whereas human infection isolates did not alter *sec* transcription levels. The influence of a complex food matrix containing lactic acid such as milk was demonstrated to reduce *sec* expression, especially in late stationary phase after 48 h [54]. Other SEs, for instance *sed,* were not significantly altered under lactic acid stress [47]. How lactic acid influences toxin transcription likely depends on the genetic background of a strain. We could, however, not observe any correlation between transcription levels and factors such as clonal complex or the SEC toxin variant of the respective strain. Possibly, a larger strain set may provide further insights.

SEC protein data only partly correlated with the *sec* transcriptional patterns. Whereas BW10 showed a significant reduction in mRNA and protein levels after 10 and 24 h, SAI3 did not display any reduced transcriptional activity, but exhibited significantly reduced SEC concentrations. Especially under certain environmental stress situations such as NaCl, sorbic acid, or complex food environments, mRNA levels do not always reflect protein levels, indicating that post-transcriptional modification might be at play [55,56,57,58]. Overall, lactic acid stress led to decreased levels of SEC in all strains. Since growth under lactic acid stress was similar to control conditions, factors independent of growth rates must be at play. Again, we did not observe any correlation between toxin production and clonal complex, source, or toxin variant. Still, SEC overproducers BW10 and SAI48 were also the highest SEC producers under lactic acid stress. Interestingly, previous studies reported substantially lower toxin concentrations of 1–70 ng/mL in milk and even under optimal growth conditions [55]. For SEA, an ingested amount of 60 ng was sufficient to reach SFP attack rates of almost 100%, highlighting the threat of overproducer strains such as BW10 and SAI48 [59]. Another study demonstrated even more pronounced reductions in SEA, SEB, SEC, and SED caused by lactic acid. This pronounced reduction could be expected, since higher concentrations (pH < 5) than in the present study were used [60]. In contrast, mild lactic acid stress was also reported to increase the formation of SEA [45]. It was suggested that *S. aureus* strains from human and food sources produce higher SEC levels in contrast with strains from animal sources [11]. The SEC overproducers in our study, BW10 and SAI48, originated from cases of SFP and human infection, respectively, and showed high SEC concentrations under control conditions and lactic acid stress. However, the moderate SEC producers SAR1, SAR38, and OV20 originating from bovine and ovine mastitis milk showed higher SEC concentrations than NB6 and SAI3, which originated from cases of SFP and human infection, respectively. This underlines the importance of investigating multiple strains for *S. aureus* enterotoxin analysis as postulated by previous studies [39,61,62,63]. The observed reduction in toxin expression may be magnified at lower temperatures as found in food environments.

Several SEs including SEC are regulated by the quorum-sensing system *agr*. The activity of *agr* may be influenced by other regulatory elements that react to changes in the microenvironment or external stressors. Such additional regulators may include, but are not limited to, *sigB*, *sarA*, *saeRS*, *srrAB*, *arlRS*, and *mgrA* [64,65]. pH stress was shown to affect regulatory elements in *S. aureus* such as *rot*, *agr*, and *sarA* [66]. In the presence of LAB, *agrA*, *sarA*, and *sigB* are typically downregulated, while *rot* is upregulated [48]. Lactic acid can influence toxin expression by interacting with DNA, enzymes, or structural proteins (Figure 3). Since the current data suggest a stronger decrease in SEC protein levels compared with mRNA, post-transcriptional events may be involved. The shift in internal pH may influence mRNA degradation susceptibility or interfere with translational events. For example, elongation factors were shown to be influenced by acid stress [67]. An overall decreased enzyme activity in the cytoplasm due to lower pH may impact SE handling after translation. In addition, folding and transport of SEs may be impaired by a lower pH, leading to less efficient excretion [68]. Which regulatory elements are affected remains unclear at this time.

In conclusion, our study demonstrated that lactic acid slightly decreases *sec* transcription and has a more pronounced effect on SEC protein levels. It can therefore be a useful tool in minimizing SEC synthesis during food production and preservation. Since transcription patterns and SEC concentrations are highly variable, it is of utmost importance to investigate several strains with different genomic backgrounds and isolated from different sources. We did not find any correlation between the observed data and any strain-specific properties such as toxin variant or clonal complex. Therefore, further research is needed to determine biomarkers associated with the toxicity of a strain. Future studies should also investigate the mechanistic action of lactic acid on the reduction in SEs to provide further insights into the regulation of SE production. Our research highlights the importance of food composition in mitigating SFP. Compounds such as lactic acid may be used as natural preservatives to minimize toxin production and alleviate the burden of staphylococcal intoxications.

## Figures and Tables

**Figure 1 microorganisms-09-01014-f001:**
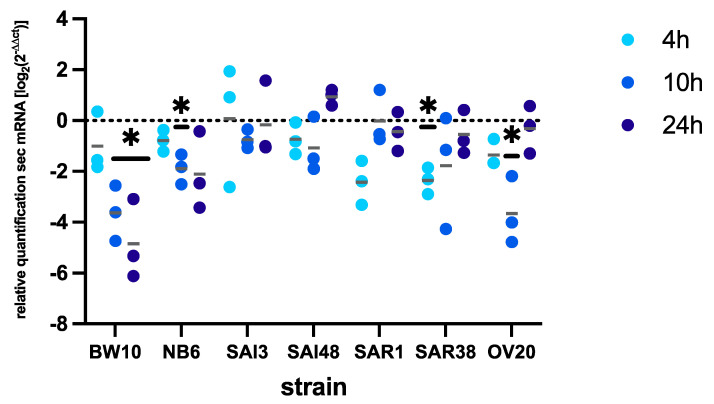
Effect of lactic acid stress on *sec* mRNA levels in *Staphylococcus aureus* strains BW10, NB6, SAI3, SAI48, SAR1, SAR38, and OV20 in exponential (4 h), early stationary (10 h), and late stationary (24 h) phase. *sec* mRNA levels are expressed as relative quantification values. *sec* mRNA expression was normalized to reference genes *rho* and *rplD* [52]. Replicates are shown as single data points; horizontal grey lines indicate means. Timepoints are signified by fill color (4 h, light blue; 10 h, medium blue; 24 h, dark blue). Significant differences in *sec* mRNA levels in LB compared with LB + lactic acid are marked by asterisks (*p* < 0.05).

**Figure 2 microorganisms-09-01014-f002:**
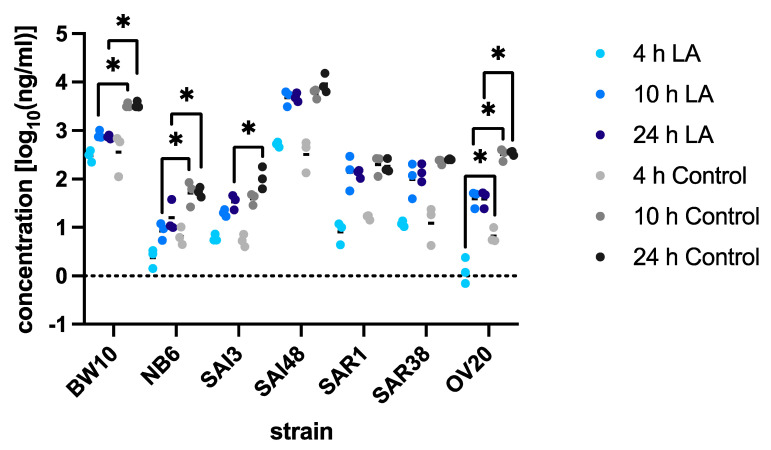
SEC concentration in log_10_ ng/mL under lactic acid stress (pH 6.0) compared with control conditions in *Staphylococcus aureus* strains BW10, NB6, SAI3, SAI48, SAR1, SAR38, and OV20 in exponential (4 h), early stationary (10 h), and late stationary (24 h) phase. SEC levels under control conditions are shown in grey; pH stress levels are shown in blue. Replicates are shown as single data points; horizontal lines indicate means. Darkening fill colors indicate progressing time points. Statistically significant differences between conditions are indicated by lines with asterisks (* *p* < 0.05).

**Figure 3 microorganisms-09-01014-f003:**
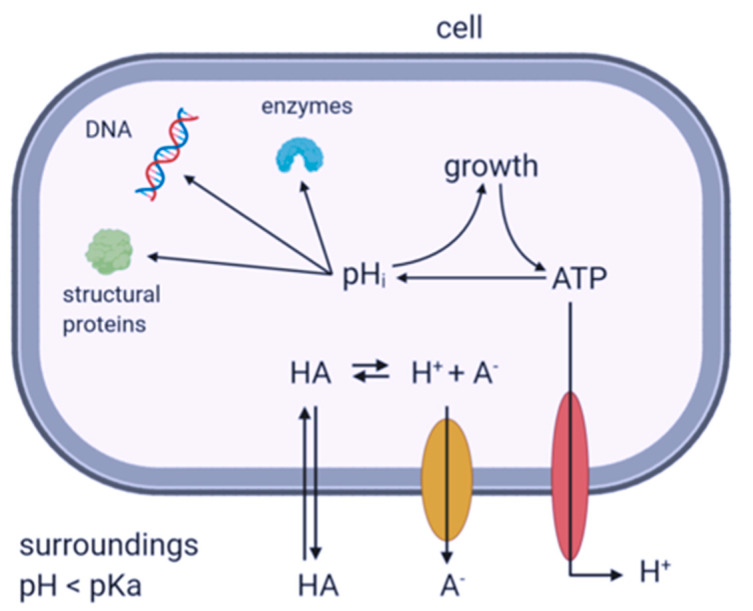
Mechanism of weak organic acids such as lactic acid on bacterial cells. The undissociated form of organic acids (HA) can cross cell membranes when the pH of the surroundings is lower than that of the cellular cytoplasm. Inside the cell, HA can dissociate and acidify the cytoplasm. Acidic pH damages or modifies internal structures such as enzymes, structural proteins, or DNA. In order to control internal pH, energy is required for active export of protons. The figure was adapted from [34] and created in BioRender.com.

**Table 1 microorganisms-09-01014-t001:** Overview of *S. aureus* strains used in this study including their SEC variants, origin, and clonal complex. ^1^ Medical Department of the German Federal Armed Forces, Germany. ^2^ Bavarian State Office of Health and Food Safety, Germany.

Strain	Protein Variant	Promotor Variant	Origin	Clonal Complex	Reference
BW10	SEC_2_	sec_p_ v1	SFP	CC45	^1^
NB6	SEC_2_	sec_p_ v1	SFP	CC45	^2^
SAI3	SEC_1_	sec_p_ v3 (H-EMRSA-15)	Human infection	CC8	[49]
SAI48	SEC_2_	sec_p_ v1 (79_S10)	Human infection	CC5	[49]
SAR1	SEC_bovine_	sec_p_ v2	bovine mastitis milk	CC151	[50]
SAR38	SEC_bovine_	sec_p_ v2	bovine mastitis milk	CC151	[50]
OV20	SEC_ovine_	sec_p_ v4	ovine	CC133	[51]

**Table 2 microorganisms-09-01014-t002:** Effect of pH stress on SEC expression. Absolute values in ng/mL including standard deviation. Effect is shown as a percentage difference under lactic acid stress (pH 6.0) compared with non-stress control conditions.

Strain	SEC Produced under pH Stress (ng/mL)									Effect of pH Stress (%)			
	4 h			10 h			24 h			4 h	10 h	24 h	Sum
BW10	307	±	81	832	±	161	740	±	69	−33	−75	−78	−186
NB6	3	±	1	9	±	3	20	±	16	−63	−85	−64	−212
SAI3	6	±	1	20	±	3	35	±	11	10	−50	−69	−109
SAI48	523	±	63	4968	±	1641	4887	±	1027	38	−17	−50	−29
SAR1	9	±	4	169	±	119	132	±	25	−46	−21	−30	−97
SAR38	12	±	2	120	±	82	143	±	59	−22	−47	−44	−113
OV20	1	±	1	41	±	15	41	±	14	−80	−87	−88	−255
SEC produced under control conditions (ng/mL)													
BW10	462	±	307	3325	±	360	3410	±	562	-	-	-	-
NB6	7	±	3	58	±	29	54	±	12	-	-	-	-
SAI3	6	±	2	40	±	10	114	±	59	-	-	-	-
SAI48	380	±	220	5988	±	1317	9868	±	4688	-	-	-	-
SAR1	16	±	2	213	±	87	189	±	65	-	-	-	-
SAR38	15	±	10	228	±	27	254	±	7	-	-	-	-
OV20	7	±	3	328	±	87	345	±	34	-	-	-	-

## Data Availability

Not applicable.

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
