# Peer review of "Mild Lactic Acid Stress Causes Strain-Dependent Reduction in SEC Protein Levels"

_microorganisms, 2021, doi:10.3390/microorganisms9051014_

Round 1

Reviewer 1 Report

The authors have made a reasonable attempt to address the critiques. There remain, however, a few items which should be corrected, some omissions that should be addressed, and choices in presenting the data that leave it difficult to interpret. Overall, it was difficult to read the revised manuscript because of the track changes, and this was probably the case for the authors as well as indicated by some errors. Specifically:

Table S1 - It was a good choice to include the plate count data rather than the spectrophotometry data and it certainly addresses many of the concerns of the (editor?) critiques about growth curves being representative of the samples that were analyzed. However the vertical dotted lines that seem to indicate sampling time points show 8 hours, rather than 10 as indicated everywhere else.

Table S2 - It is not clear what the relevance of this data is - if the goal is to show an absence of growth impairment a comparison of the treated vs. untreated would be necessary. I don't think such an analysis is required, the figure is sufficient.

L162 - Log10 transformed - typically gene expression data is log2 transformed, doing that here with labeling on the axis would make interpreting these data substantially easier.

L185 - refers to fig 1 B & C, but figure 1 no longer has a panel C.

Figure 1 & 2 - the box and whisker plots are an odd choice for data that is composed of only 6 (fig 1) and 2 (fig 2) values. The presentation in fig 1B of the individual values is much easier to interpret.

Figure 1 - it is not clear how this analysis was conducted. The legend refers to normalizing to rho and rplD but how? rRNA and the housekeeping sigma factor are more typically used for normalization, both rho and rplD may be regulated under conditions relevant to this work. Do the six points in panel B represent biological triplicates of sec normalized to each of these? If so which points are for which denominator? The data presented in panel A is probably sufficient here, but would be better as individual values plotted on a log2 scale.

Figure 2 - agree with the critique that levels of significance don't add anything.

Figure S3 - corcles -> circles

Author Response

The authors have made a reasonable attempt to address the critiques. There remain, however, a few items which should be corrected, some omissions that should be addressed, and choices in presenting the data that leave it difficult to interpret. Overall, it was difficult to read the revised manuscript because of the track changes, and this was probably the case for the authors as well as indicated by some errors. Specifically:

  1. Table S1 - It was a good choice to include the plate count data rather than the spectrophotometry data and it certainly addresses many of the concerns of the (editor?) critiques about growth curves being representative of the samples that were analyzed. However the vertical dotted lines that seem to indicate sampling time points show 8 hours, rather than 10 as indicated everywhere else.

The mistake in the graphs was corrected so that now the sampling point at 10h is shown.

  1. Table S2 - It is not clear what the relevance of this data is - if the goal is to show an absence of growth impairment a comparison of the treated vs. untreated would be necessary. I don't think such an analysis is required, the figure is sufficient.

The table was removed from the supplementary material as requested.

  1. L162 - Log10 transformed - typically gene expression data is log2 transformed, doing that here with labeling on the axis would make interpreting these data substantially easier.

Log10 transformation was done to ensure normal distribution of the data in order to correctly perform all statistical analyses. We have now plotted the values on a log2 scale to facilitate the graphical interpretation. In addition, the boxplots were replaced by individual values as in panel 1B.

  1. L185 - refers to fig 1 B & C, but figure 1 no longer has a panel C.

The reference was corrected to refer to figure 1.

  1. Figure 1 & 2 - the box and whisker plots are an odd choice for data that is composed of only 6 (fig 1) and 2 (fig 2) values. The presentation in fig 1B of the individual values is much easier to interpret.

The boxplots have been replaced by plots depicting individual values.

  1. Figure 1 - it is not clear how this analysis was conducted. The legend refers to normalizing to rho and rplD but how? rRNA and the housekeeping sigma factor are more typically used for normalization, both rho and rplD may be regulated under conditions relevant to this work. Do the six points in panel B represent biological triplicates of sec normalized to each of these? If so which points are for which denominator? The data presented in panel A is probably sufficient here, but would be better as individual values plotted on a log2 scale.

Reference genes rho and rplD have been validated under lactic acid stress and were shown to be stably expressed under the present stress conditions by Sihto et al., 2014 (https://academic.oup.com/femsle/article/356/1/134/637478). The respective reference was included in the figure legend for further clarification.

The formula for to calculate expression values was now included in the materials and methods section to further clarify how normalization was done in line 119:

The values from both reference genes were now combined to form a single data point in figure 1. Values are now plotted on a log2 scale.

Panel B was removed, as suggested.

  1. Figure 2 - agree with the critique that levels of significance don't add anything.

The plot has been adapted so that only a single level of significance is shown.

  1. Figure S3 - corcles -> circles

The typo was corrected.

Reviewer 2 Report

In this manuscript by Etter et al., the effects on the production of the superantigen SEC in S.aureus by lactic acid stress is investigated. The manuscript is well-written and easy to read. The authors have analysed the mRNA expression levels as well as the protein expression levels of SEC in different strains with and without lactic acids stress. The overall conclusion from the authors is that there is no major reduction of SEC after lactic stress on the mRNA level, but larger effects are seen for some strains at the protein level, which the authors explain by effects on post transcriptional modifications.

Comments:

  1. For the mRNA levels the authors normalise the data to internal controls (rho and rplD), it is not clear to me if the protein levels were normalised? If they were not, they should at least be compared to some internal control protein to exclude that the reduction is not a general phenomenon caused by the different growth conditions.

  1. As the authors suggest that post transcriptional events may explain the change in protein levels after stress, they should extend this discussion and be clearer on what they are referring to and how it may affect the translation of the enterotoxins. Potentially, this should also be confirmed by experiments.

Author Response

In this manuscript by Etter et al., the effects on the production of the superantigen SEC in S.aureus by lactic acid stress is investigated. The manuscript is well-written and easy to read. The authors have analysed the mRNA expression levels as well as the protein expression levels of SEC in different strains with and without lactic acids stress. The overall conclusion from the authors is that there is no major reduction of SEC after lactic stress on the mRNA level, but larger effects are seen for some strains at the protein level, which the authors explain by effects on post transcriptional modifications.

Comments:

  1. For the mRNA levels the authors normalise the data to internal controls (rho and rplD), it is not clear to me if the protein levels were normalised? If they were not, they should at least be compared to some internal control protein to exclude that the reduction is not a general phenomenon caused by the different growth conditions.

Since the antibodies in our ELISA assay are specific to SEC we did not include any other proteins in the analysis. As previously shown by Poli et al., 2002 the assay buffer itself does not influence the outcome of this ELISA assay. In addition, since mRNA toxin levels and especially the amount of transcribed reference genes rho and rplD did not substantially differ under stress conditions we do not think that there is a global downregulation of all proteins under lactic acid stress. A standard curve was used to perform absolute quantification of the protein concentration in the supernatant. All supernatants were taken at the same time as our mRNA samples and are therefore standardized to bacterial growth. The procedure was performed according to previously published ELISAs with enterotoxins (Poli et al., 2002; Wallin-Carlquist et al., 2010).

  1. As the authors suggest that post transcriptional events may explain the change in protein levels after stress, they should extend this discussion and be clearer on what they are referring to and how it may affect the translation of the enterotoxins. Potentially, this should also be confirmed by experiments.

We did extend the discussion to include a more thorough investigation of which post-transcriptional events may be affected in lines 246-257:

“Since the current data suggests a stronger decrease in SEC protein levels compared to mRNA, post-transcriptional events may be involved. The shift in internal pH could influence mRNA degradation susceptibility or interfere with translational events. For example, elongation factors were shown to be influenced by acid stress [67]. An overall decreased enzyme activity in the cytoplasm due to lower pH could impact SE handling after translation. In addition, folding and transport of SEs could be impaired by a lower pH, leading to less efficient excretion [68]. Which regulatory elements are affected remains unclear at this time.”

Reviewer 3 Report

  1. The title may need some revision.
  2. Line #9- Please rephrase to use a more appropriate choice word as a major cause of instead of important.
  3. Rest of the manuscript looks good just needs english grammar check.

Author Response

  1. The title may need some revision.

The title was changed to: Mild lactic acid stress causes strain dependent reduction in SEC protein levels

  1. Line #9- Please rephrase to use a more appropriate choice word as a major cause of instead of important.

The sentence has been rephrased as suggested.

  1. Rest of the manuscript looks good just needs english grammar check.

Grammar and spelling have been thoroughly checked and occasional typos were corrected

Round 2

Reviewer 2 Report

The extended paragraph to my second comment is fine, however I do not think the authors have fulfilled my first requirement, where I state that "..., they should at least be compared to some internal control protein to exclude that the reduction is not a general phenomenon caused by the different growth conditions".

Round 3

Reviewer 2 Report

No further comments.

This manuscript is a resubmission of an earlier submission. The following is a list of the peer review reports and author responses from that submission.

Round 1

Reviewer 1 Report

Staphylococcal enterotoxins are involved in food poisoning and bovine mastitis. Lactic acid bacteria are a counter measure to both problems, but how they influence enterotoxin expression is unknown. In this manuscript, the authors report a study of the influence of lactic acid on the expression of staphylococcal enterotoxin C, at the mRNA and protein level, in a panel of clinical and veterinary isolates. Despite substantial variability between strains, in most strains they find a small decrease in mRNA levels but a larger decrease in protein levels. The results suggest post-transcriptional regulation may be important in determining sec levels. The paper is well-written, addresses an open question, and uses appropriate techniques.

A few minor revisions could improve the manuscript:

L152-4 It is not clear where the (-29%) and (-255%) came from - they are not present in table 2.

Figure 2A is redundant, Figure 2B should be sufficient

There is a paper that describes the effect of LAB on sec expression that is not referenced but should be, either in the introduction or the discussion, it is PMID 27342241.

The supplementary table could be moved to the main section.

Author Response

We thank the reviewers for their examination of the manuscript. We addressed all remarks in a point-by-point response and revised parts of the article as recommended. The provided feedback has helped us to improve the manuscript and we hope that it is now fit for publication.

L152-4 It is not clear where the (-29%) and (-255%) came from - they are not present in table 2.

Thank you for pointing out the lack of clarity here. We have included the respective information in table 2.

Figure 2A is redundant, Figure 2B should be sufficient

            We have removed figure 2A and left only figure 2B for a clearer representation of the data.

There is a paper that describes the effect of LAB on sec expression that is not referenced but should be, either in the introduction or the discussion, it is PMID 27342241.

Thank you for bringing this publication to our attention. We have included it in the text to provide further background information on the influence of LAB on SEC expression in lines 48-49.

The supplementary table could be moved to the main section.

We have included the additional information from the supplementary table S1 in table 2 and removed the supplementary material.

Reviewer 2 Report

The role of SEC on staphylococcal infective endocarditis is not formally established. A preprint manuscript should not be referenced.

Lactic acid could suppress sec mRNA and protein levels in some strains, but the role of lactic acid as a suppressor of SEC production is not conclusive.

The authors published an article in Toxins about sec, and this study adds little value to their former study.

It does not seem to merit from publishing this study without the investigation in the background mechanism of lactic acid on suppression of SEC production.

Author Response

We thank the reviewers for their examination of the manuscript. We addressed all remarks in a point-by-point response and revised parts of the article as recommended. The provided feedback has helped us to improve the manuscript and we hope that it is now fit for publication.

The role of SEC on staphylococcal infective endocarditis is not formally established. A preprint manuscript should not be referenced.

We have removed the claim and the corresponding reference as suggested.

Lactic acid could suppress sec mRNA and protein levels in some strains, but the role of lactic acid as a suppressor of SEC production is not conclusive.

We agree that the rather small reduction in mRNA levels and the strain variability of the investigated strain set might not support the claim on its own. But we are convinced that the substantial reduction of SEC measured in the ELISA assay does indeed demonstrate the potential of lactic acid in reducing enterotoxin expression. We rephrased the conclusion in lines 231-232 for clarification.

The authors published an article in Toxins about sec, and this study adds little value to their former study.

We have recently published a review on SEC in Toxins. There, we summarized the current knowledge on SEC and its role in disease. In this review, one of the main features of SEC that we pointed out was the diversity of toxin variants and the lack of evidence between environmental stresses and their role in the expression of different SEC variants. In the current original article, we present novel data that can make an important contribution towards filling this gap.

It does not seem to merit from publishing this study without the investigation in the background mechanism of lactic acid on suppression of SEC production.

While it would be interesting to elucidate the mechanistic action of lactic acid on toxin production, this would substantially exceed the scope of this study. However, we are looking forward to investigate this aspect in more detail in the future. We have added a respective paragraph to the manuscript in lines 239-240 to highlight some of the limitations of the study and to present an appropriate outlook for the future.
